# ADAPTIVE N-STEP BOOTSTRAPPING WITH OFF-POLICY DATA

## ABSTRACT

The definition of the update target is a crucial design choice in reinforcement learning. Due to the low computation cost and empirical high performance, n-step returns with off-policy data is a widely used update target to bootstrap from scratch. A critical issue of applying n-step returns is to identify the optimal value of $n$. In practice, $n$ is often set to a fixed value, which is either determined by an empirical guess or by some hyper-parameter search. In this work, we point out that the optimal value of $n$ actually differs on each data point, while the fixed value $n$ is a rough average of them. The estimation error can be decomposed into two sources, off-policy bias and approximation error, and the fixed value of $n$ is a trade-off between them. Based on that observation, we introduce a new metric, **policy age**, to quantify the off-policyness of each data point. We propose the **Adaptive N-step Bootstrapping**, which calculates the value of $n$ for each data point by its policy age instead of the empirical guess. We conduct experiments on both MuJoCo and Atari games. The results show that adaptive n-step bootstrapping achieves state-of-the-art performance in terms of both final reward and data efficiency.

## 1 INTRODUCTION

The goal of reinforcement learning (RL) is to find an optimal policy by interacting with the environment. In order to do that, a RL algorithm needs to define a target, e.g., Q function or value function, and update it iteratively to bootstrap from scratch. The challenge of designing an efficient update target manifests both on the sample complexity and computation complexity. Ideally, the target should only be updated by the data generated by the corresponding policy to obtain an unbiased estimate (Sutton & Barto, 2018), while the amount of the data needs to reach a certain scale to control the variance (Schulman et al., 2015). These two requirements limit the update frequency and lead to a high sample complexity finally. On the computational part, the consideration is to make a trade-off between the computation cost of each step and the number of total steps. Monte-Carlo returns has advantages on generalization (behave well with function approximation) and exploration (a quick propagation of new findings) at the cost of computing the whole trajectory on each step (Sutton & Barto, 2018). Bootstrapping methods apply readily on off-policy data and control the trace length (Sutton & Barto, 2018). However, they require more update steps to converge comparing with Monte-Carlo returns. Those design concerns are nested together, which makes it hard to achieve a good balance.

N-step returns (Sutton & Barto, 2018) serves as the basis of various update targets, due to its flexibility and simple implementation. Together with off-policy learning (Sutton & Barto, 2018) and the replay buffer (Mnih et al., 2015), n-step returns is able to update the target frequently while ensures that the variance is in a controllable range. However, a systematical study (Fedus et al., 2020) reveals that the performance of n-step returns highly relies on the exact value of $n$. Since the underlying working mechanism is unclear, previous research can only give some vague suggestion based on empirical results, that simply increases the value from one to a larger number, e.g. 3 or 4.

In this paper, we illustrate that the estimation error of n-step returns can be decomposed into off-policy bias (under-estimation part) and approximation error (over-estimation part), and the selection of $n$ controls the balance between them. Data stored in the replay buffer are generated by previous policies. Thus, adopting them for update introduces the off-policy bias. Since the current policy

is better than previous policies, the off-policy bias is an underestimation. Replay buffer is not the only source of the off-policy bias, epsilon-greedy exploration also introduces the off-policy issue. On the other hand, n-step returns adopts a max operator explicitly (on Q-learning based algorithms) or implicitly (on actor-critic algorithms) on an existing function to approximate the real target. The max operator brings the approximation error, which is an overestimation. Sec 4 gives the formal definition of the decomposition and verifies the conclusion by experiments. According to our analysis, the quantity of the off-policy bias and approximation error varies a lot on different data points. Thus, a fixed value of $n$ is just a rough average, and there is plenty of room for improvement.

We introduce a new metric, **policy age**, to quantify the off-policyness of each data point. As the policy age grows, the off-policy bias increases linearly, while the approximation error decreases exponentially. Based on this observation, we propose a novel algorithm, named *adaptive n-step bootstrapping*. Given the policy age of each data point, adaptive n-step calculates the optimal $n$ by an exponential function. Hyperparameter of the function is determined by the tree-structured parzen estimator (Bergstra et al., 2011). We conduct extensive experiments on both MuJoCo and Atari games. Adaptive n-step bootstrapping outperforms all fixed-value $n$ settings with a large margin, in terms of both data efficiency and final reward. For the other update target definitions, we select Retrace (Munos et al., 2016) as a representative. Compared with Retrace, our method maintains the performance advantage under the premise of low computational complexity and simple implementation.

## 2 RELATED WORKS

### 2.1 RESEARCH ON N-STEP RETURNS

The recent works on n-step returns focus on finding the optimal value of $n$. In Ape-X (Horgan et al., 2018) and R2D2 (Kapturowski et al., 2019), the value of $n$ is fixed, which is set by manual tuning or hyper-parameter search. Rainbow (Hessel et al., 2018) figures out that the final performance is sensitive to the value of $n$, and $n = 3$ achieves the best score in most cases on Atari games. Fedus et al. (2020) verifies that setting $n$ to 3 is a good choice, and further reveals that the replay buffer must also be large enough to gain performance benefits. Those researches give some heuristic rules of setting the value of $n$, but the underlying mechanism of why $n = 3$ performs better than one-step temporal difference is still unclear.

### 2.2 OTHER UPDATE TARGETS

To improve the performance of the vanilla n-step returns, there are many other update target definitions in the literature (Hernandez-Garcia & Sutton, 2019). Importance sampling (IS) (Precup et al., 2000) provides a simple way to correct the off-policy bias. It can be seen as a weighted average of multiple one-step TD(0) target. However, IS brings large (and possibly infinite) variance, which makes it impractical on large-scale problems.

Retrace (Munos et al., 2016) clips the IS ratio to a maximum value of 1 to reduce the large variance of IS targets. It has many applications in recent reinforcement learning agents, like distributed off-policy learning agent Reactor (Gruslys et al., 2017). The most serious disadvantage of Retrace is its high computation cost. Retrace needs to calculate $O(n)$ times of $Q$ and $O(n)$ times of $\pi$ in each time, compared with only $O(1)$ from n-step returns ($n$ is trace length). In large-scale problems, evaluating $Q$ and $\pi$ requires a forward pass in the neural network, which is slow and expensive. Reactor (Gruslys et al., 2017) calculates the Retrace target as a linear combination of many n-step targets and dispatches those calculation workloads into different nodes for acceleration. Since the computation complexity is still high, reactor can not work under limited resources. Furthermore, even without considering the calculation cost, the application scope of Retrace is not as good as n-step returns, as reported in Hernandez-Garcia & Sutton (2019).

## 3 PRELIMINARIES

Reinforcement learning's goal is to find an optimal policy $\pi^*$ with maximal discounted returns $R_\pi = \mathbb{E}_\pi[\sum_t \gamma^{t-1} r_t]$ given the Markov Decision Process (MDP). To achieve this, agents often

estimate the state-action value function $q_\pi(s, a) = \mathbb{E}_\pi[\sum_t \gamma^{t-1} r_t | s_0 = s, a_0 = a]$. Let $Q_\pi(s, a)$ denote the estimation of $q_\pi(s, a)$. In tabular settings, $Q_\pi$ can be represented by a table of all state-action pairs $\langle s, a \rangle$, while in large-scale settings, $Q_\pi$ is often approximated by a deep neural network (DNN) with parameter $\theta$, written as $Q_{\pi;\theta}$.

During the training process, $Q_\pi$ is continuously updated by the update target $\hat{G}_\pi$, which is calculated from data points $(s, a, r, s')$. In tabular settings (Watkins & Dayan, 1992), the update equation can be written as: $Q_\pi(s, a) \leftarrow Q_\pi(s, a) + \alpha[\hat{G}_\pi(s, a) - Q_\pi(s, a)]$, where $\alpha$ is the learning rate. In large-scale settings (Mnih et al., 2015; Lillicrap et al., 2016), $Q_{\pi;\theta}$ is updated by mini-batch gradient descent on the neural network parameter $\theta$ as: $\theta \leftarrow \theta - \alpha \frac{1}{N} \sum_{i=1}^{N} \nabla_\theta L(Q_{\pi;\theta}(s_i, a_i), \hat{G}_\pi(s_i, a_i))$, where $L$ is the loss function.

Off-policy learning adopts two policies, the behavior policy $\mu$ for generating data points and the target policy $\pi$ for learning from the data points. Replay buffer (Fedus et al., 2020) is often used together with the off-policy to handle the increasing sample complexity brought by large-scale problems. Agents draw data points from the replay buffer to update the estimator $Q_\pi$. This enables agents to learn from past experiences, thus yields higher sample efficiency. In the sequel, the term **off-policy learning** refers to adopt off-policy learning together with the replay buffer, as used by most recent off-policy algorithms.

N-step returns (Sutton & Barto, 2018) together with off-policy learning have strong empirical performance (Hessel et al., 2018), as well as being easy to calculate. It calculates the $\hat{G}_\pi^n$ as:

$$\hat{G}_\pi^n(s, a) = r_0 + \gamma r_1 + \cdots + \gamma^{n-1} r_{n-1} + \gamma^n \mathbb{E}_{a_n \sim \pi(s_n)}[Q_\pi(s_n, a_n)]. \tag{1}$$

It can be seen as a mix of Monte-Carlo (MC) estimation of $q_\pi \approx \sum_t \gamma^{t-1} r_t$ ($n \to \infty$ case) and one-step TD(0) estimation $r_0 + \gamma \mathbb{E}_{a \sim \pi(s_1)}[Q_\pi(s_1, a)]$ ($n = 1$ case). In off-policy learning, to calculate n-step returns $\hat{G}_\pi^n(s_0, a_0)$ for data point $(s_0, a_0, r_0, s'_0)$, we draw consecutive transitions $(s_t, a_t, r_t, s'_t)_{t=0,1,2,...}$ in the same trajectory $\tau$ as current data point $(s_0, a_0, r_0, s'_0)$ from replay buffer. As n-step returns estimates $q_\pi$ using trajectory $\tau$ generated by behavior policy $\mu$, the discrepancy between $\pi$ and $\mu$ makes it biased. We define this off-policy induced bias as **off-policy bias**, and difference between $\pi$ and $\mu$ as **off-policyness**.

# 4 UNDERLYING WORKING MECHANISM OF N-STEP BOOTSTRAPPING

N-step returns lays in the center of designing the update target. It not only unifies the Monte-Carlo returns and one-step temporal difference but also lays the foundation of the eligibility traces (Singh & Sutton, 1996). Together with off-policy learning, n-step bootstrapping works well because it achieves a good balance between the bias (TD) and variance (MC). Considering its importance and wide application, the underlying mechanism of n-step returns has not been studied in detail. In this section, we give a careful analysis of why n-step bootstrapping works and what property the optimal selection of $n$ should satisfy.

## 4.1 DECOMPOSITION OF THE ESTIMATION ERROR

To understand how n-step bootstrapping works, we conduct a systematical analysis of the estimation error. We formalize the estimation error of an update target $\hat{G}_\pi$ as its difference with ground truth $q_\pi$ for every $(s, a)$ pair that the agent experienced:

$$\mathcal{E}(\hat{G}_\pi) = \mathbb{E}_\mu[\hat{G}_\pi(s, a) - q_\pi(s, a)]. \tag{2}$$

As shown in Eq. 1, the estimation error consists of two parts — the **off-policy bias** and **approximation error**. The off-policy bias is introduced by the difference of $\pi$ and $\mu$, as we mentioned before, while the approximation error comes as agent's estimation of $Q_\pi$ is not perfect, e.g. $Q_\pi \neq q_\pi$. Note that, the behavior policy $\mu$ is an older version of the target policy $\pi$.

We split these two types of error by defining two intermediate targets, $\hat{G}_{\pi,q_\pi;\tau}^n$ and $\hat{G}_{\pi,Q_\pi;\tilde{\tau}}^n$, to eliminate the other type of error.

$\hat{G}_{\pi,q_{\pi};\tau}^{n}$ adopts the ground truth $q_{\pi}$ instead of the estimated $Q_{\pi}$ to eliminate approximation error:

$$\hat{G}_{\pi,q_{\pi};\tau}^{n}(s,a) = r_0 + \gamma r_1 + \cdots + \gamma^{n-1} r_{n-1} + \gamma^n \mathbb{E}_{a_n \sim \pi(s_n)}[q_{\pi}(s_n, a_n)].$$

$\hat{G}_{\pi,Q_{\pi};\tilde{\tau}}^{n}$ uses the trajectory $\tilde{\tau} = (\tilde{s}_t, \tilde{a}_t, \tilde{r}_t, \tilde{s}_t')_{t=0,1,2,\ldots}$ which is generated by the current policy $\pi$ instead of the old trajectory $\tau$ to remove the off-policy bias:

$$\hat{G}_{\pi,Q_{\pi};\tilde{\tau}}^{n}(s,a) = \tilde{r}_0 + \gamma \tilde{r}_1 + \cdots + \gamma^{n-1} \tilde{r}_{n-1} + \gamma^n \mathbb{E}_{\tilde{a}_n \sim \pi(\tilde{s}_n)}[Q_{\pi}(\tilde{s}_n, \tilde{a}_n)].$$

Then we can quantify the off-policy bias $\mathcal{E}_{\text{offpolicy}}$ and the approximation error $\mathcal{E}_{\text{approx}}$ independently as:

$$\mathcal{E}_{\text{offpolicy}}(\hat{G}_{\pi}^n) = \mathcal{E}(\hat{G}_{\pi,q_{\pi};\tau}^{n}), \quad \mathcal{E}_{\text{approx}}(\hat{G}_{\pi}^n) = \mathcal{E}(\hat{G}_{\pi,Q_{\pi};\tilde{\tau}}^{n}).$$

Now, we can decompose the total error $\mathcal{E}(\hat{G}_{\pi}^n)$ into the sum of off-policy bias and approximation error, plus a negligible small residual term $\mathcal{E}_{\text{residual}}(\hat{G}_{\pi}^n)$:

$$\begin{aligned} \mathcal{E}_{\text{residual}}(\hat{G}_{\pi}^n) &= \mathcal{E}(\hat{G}_{\pi}^n) - \mathcal{E}_{\text{offpolicy}}(\hat{G}_{\pi}^n) - \mathcal{E}_{\text{approx}}(\hat{G}_{\pi}^n) \\ &= \gamma^n (\mathbb{E}_{a \sim \pi(s_n)}[Q_{\pi}(s_n, a) - q_{\pi}(s_n, a)] - \mathbb{E}_{\tilde{a} \sim \pi(\tilde{s}_n)}[Q_{\pi}(\tilde{s}_n, \tilde{a}) - q_{\pi}(\tilde{s}_n, \tilde{a})]). \end{aligned}$$

The residual term $\mathcal{E}_{\text{residual}}(\hat{G}_{\pi}^n)$ is a difference of the discounted error $\gamma^n \mathbb{E}_{a \sim \pi(s)}[Q_{\pi}(s,a) - q_{\pi}(s,a)]$ on terminal states $s_n$ and $\tilde{s}_n$. If $n$ is small, the difference between $Q_{\pi}$ and $q_{\pi}$ on $s_n$ will be close to the difference between $Q_{\pi}$ and $q_{\pi}$ on $\tilde{s}_n$. Otherwise, the discount factor $\gamma^n$ will shrink exponentially, making the residual term very small.

The experimental results in Sec 4.2 show that the residual term is an order of magnitude smaller than the two main sources in practice. Thus, it can be ignored and we get an approximate decomposition:

$$\mathcal{E}(\hat{G}_{\pi}^n) \approx \mathcal{E}_{\text{offpolicy}}(\hat{G}_{\pi}^n) + \mathcal{E}_{\text{approx}}(\hat{G}_{\pi}^n). \tag{3}$$

### 4.2 VERIFICATION BY EXPERIMENTS

In this section, we perform tabular Q-Learning and Soft Actor Critic (Haarnoja et al., 2018a) on the Pendulum-v0 task (Brockman et al., 2016) to verify the decomposition quantitatively. For tabular Q-learning, we added replay buffer (Mnih et al., 2015) as recent off-policy learning does, and discretization to deal with continuous observation space and action space.

The approximation error is an overestimation, which means $\mathcal{E}_{\text{approx}} \geq 0$, while the off-policy bias is an underestimation, $\mathcal{E}_{\text{offpolicy}} \leq 0$. Single-step temporal difference target ($n = 1$) will have a large approximation error, while too many steps ($n$ is large) leads to too much underestimation that cannot be balanced out. **N-step returns works because a suitable selection of $n$ makes the overestimation and underestimation cancel each other.**

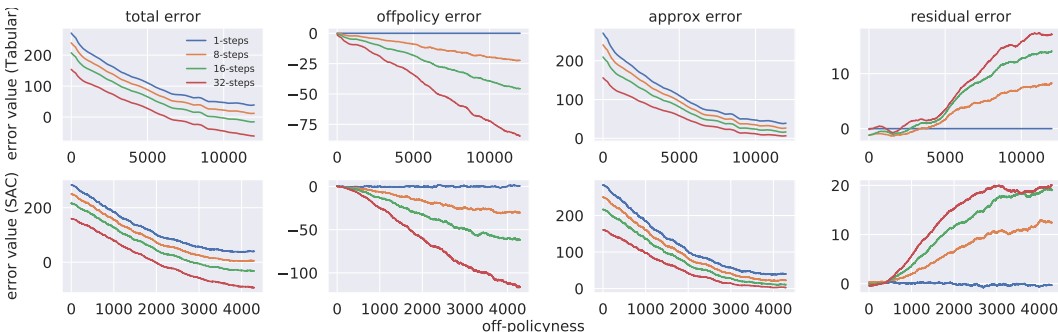

Figure 1: N-step target errors. Top row is tabular settings, bottom is SAC. X-axis is off-policyness, quantified by policy age that is described in the beginning of Sec 5.

Overestimation of $\mathcal{E}_{\mathrm{approx}}$ is a well-studied issue. The greedy policy $\pi$ acts as a max operator in Q-learning targets. Gradient ascent on $\pi$ in actor-critic architectures is also an implicit max operator (Fujimoto et al., 2018). The max operator is one main cause of overestimation, as reported in (Hasselt, 2010). Usage of function approximation (Thrun & Schwartz, 1999; Van Hasselt et al., 2015) in large-scale settings also boosts this issue. Off-policy learning with replay buffer is the root cause of the underestimation of $\mathcal{E}_{\mathrm{offpolicy}}$. With less learning update iterations, $\mu$ often has worse performance than $\pi$. Estimating $q_\pi$ using $\mu$ leads to underestimating, as we are accumulating $\mu$'s reward in $\hat{G}_\pi^n$.

As shown in Figure 1, on per-datapoint perspective, off-policy bias $\mathcal{E}_{\mathrm{offpolicy}}$ grows with off-policyness. Older data points take a large proportion in replay buffer, but data points from current policy $\pi$ have only a few, so $Q_\pi$ has been updated with much more older data points. That makes approximation error $\mathcal{E}_{\mathrm{approx}}$ decrease with off-policyness.

On value of $n$ perspective, larger $n$ leads to less approximation error, as weight $\gamma^n$ of estimated $Q_\pi$ shrinks exponentially. But larger $n$ also leads target $\hat{G}_\pi^n$ accumulating more rewards from $\mu$, thus enlarges off-policy bias.

## 5 IDENTIFY THE OPTIMAL VALUE OF N

The magnitude of off-policy bias is closely related to off-policyness. To analyze the error of n-step returns, we need a precise measurement of the off-policyness. Evaluating real off-policyness directly, e.g. calculating the real difference between $\pi$ and $\mu$ is hard, as calculating the difference of two policies is non-trivial and computational heavy, especially for complicated environments. So we introduce a new metric **policy age**, which is simple to evaluate and also predicts well of real off-policyness. For every data point $(s_t, a_t, r_t, s_t')$, policy age is defined as number of update steps between $\pi$ and $\mu$. As we show in Figure 2, policy age predicts accurately of the difference between $\pi$ and $\mu$, e.g. $\mathbb{E}_\mu[|\log \pi(a|s) - \log \mu(a|s)|]$ for every $(s, a)$ pair that agent experienced. We will use policy age as an indicator of off-policyness in the paper.

### 5.1 ADAPTIVE N-STEP BOOTSTRAPPING

As we pointed out in Sec 4.2, the optimal value of $n$ achieves a balance between overestimation and underestimation. Since the optimal value of $n$ varies by policy age, a fixed $n$ value is only a coarse approximation. For each data point, the selection of $n$ should be calculated individually to achieve the best performance.

We propose a novel algorithm, **Adaptive N-step Bootstrapping**, to select the optimal $n$ which achieves minimal error for every data point. We define error for policy age $p$ as:

$$\mathcal{E}^p(\hat{G}_\pi) = \mathbb{E}_{\mu_p}[\hat{G}_\pi(s, a) - q_\pi(s, a)], \tag{4}$$

where $\mu_p$ refers to behavior policy that is $p$ updates away from $\pi$. Then, the optimal $n^*$ for every data point is calculated as:

$$n^*(p) = \arg\min_n |\mathcal{E}^p(\hat{G}_\pi^n)|. \tag{5}$$

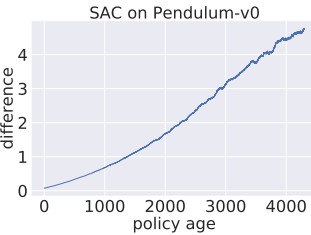

Figure 2: The x axis is the policy age, and the y axis is the difference between $\mu$ and $\pi$, which grows approximately linearly as the policy age grows.

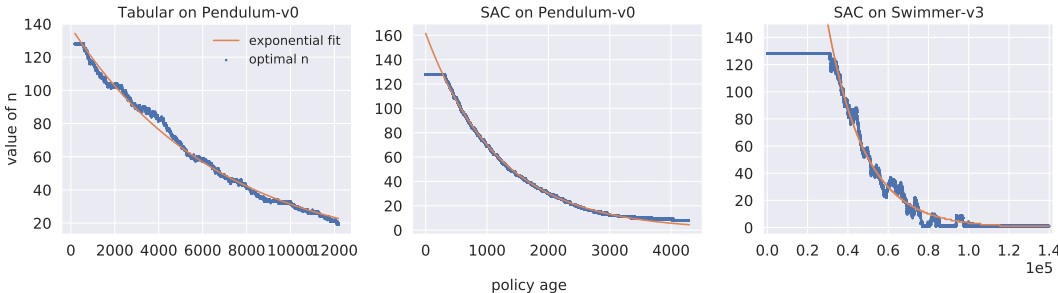

Figure 3: $n^*$ and exponential fitting. Note that we only evaluate error for 128 steps, so the value of $n^*$ may be clipped to a maximum of 128.

As calculating error $\mathcal{E}^p(\hat{G}_\pi^n)$ requires the ground truth $q_\pi$, directly solving Eq 5 on every data point is very expensive on large-scale environments. We start by solving $n^*$ for a small number of data points. As shown in Figure 3, these data points reveal that $n^*(p)$ is an exponential form in a wide variety of settings, both tabular and large-scale.

This exponential pattern is rooted in the management of replay buffer and mini-batch gradient descent. In replay buffer, the larger the policy age of a data point, the more times that it will be sampled into the mini-batch. This difference on sample times inflects both the off-policy bias and the approximation error. The off-policy bias part is simple, it increases linearly as the policy age grows. The approximation error is a little bit more complicated because the sample data update the parameter through a gradient descent. Bhandari et al. (2018) describes a similar condition, and it concludes that the rate of convergence is exponential. In summary, as the policy age grows, the off-policy bias increases linearly while the approximation error decreases exponentially. Thus, we use an exponential approximation for $n^*$ as follows:

$$n^*(p) = \arg\min_n |\mathcal{E}^p(\hat{G}_\pi^n)| \tag{6}$$

$$\approx \text{round}(n_{\max} * e^{-\log(n) \cdot \min(1, \frac{p}{d})}), \tag{7}$$

where $p$ is the policy age for the data point. The maximum factor $n_{\max}$ and decay rate $d$ are hyperparameters.

Adaptive n-step returns can also be seen as a form of off-policy correction, cutting trace when the difference between $\pi$ and $\mu$ is large. It is simpler and more stable than IS and Retrace because it does not rely on the metric like IS ratio, which may have infinite variance. Algorithm 5.1 describes how adaptive n-step bootstrapping works.

---

**Algorithm 1** Adaptive N-step Bootstrapping

**Hyperparameter:** Maximum steps $n_{\max}$, decay rate $d$
**for** each bootstrapping iteration **do**
    sample a batch $(s_i, a_i, r_i, s_i')$ from replay buffer
    **for** each data point $i$ in batch **do**
        calculate policy age $p$ for data point $i$
        $n \leftarrow \text{round}(n_{\max} * e^{-\log(n_{\max}) \cdot \min(1, \frac{p}{d})})$
        $g_i \leftarrow \hat{G}_i^n(s_i, a_i)$
    **end for**
    update $Q_\pi$ using batch $(s_i, a_i, r_i, s_i')$ with target $g_i$
**end for**

---

## 5.2 HYPERPARAMETER SELECTION

As we show in Sec 5.1, solving Eq 5 on limited data points is enough to reveal the exponential form. $n^*$ of complicated Swimmer-v3 environment is also successfully reconstructed.

We also adopt the same setting to solve the optimal value of $n_{\max}$ and $d$. The exact solution on Swimmer-v3 give us $n_{\max} = 755$. On near-onpolicy case, adaptive n-step behaves like MC target, which is unbiased for on-policy case (Sutton & Barto, 2018). However, MC target also has a large variance, which leads to performance degradation. So we need variance control measurements, like the $\lambda$ factor in Retrace (Munos et al., 2016). Vanilla n-step returns sets $n$ to a small value to do variance control (Sutton & Barto, 2018). We clip $n_{\max}$ to reduce the variance in our method.

The notorious bias-variance trade-off makes it non-trivial to solve optimal $n_{\max}$. However, the solved $d = 122952$ is very close to $d = 100000$ that works for all MuJoCo tasks, hinting the best hyperparameter range. We found that the best hyperparameter varies little with different environments. With this assumption, we use the tree-structured parzen estimator (Bergstra et al., 2011) to optimize hyperparameter on one environment and use this single set of hyperparameter for the whole benchmark.

## 6 EXPERIMENTAL RESULTS

Q-learning and Actor-Critic methods span the whole space of model-free reinforcement learning algorithms. Q-learning updates the state-action value function $Q_\pi$ and select action with maximum $Q_\pi$ in $\epsilon$-greedy manner. Actor-Critic has an explicit representation of $\pi$, and update $Q_\pi$ and $\pi$ simultaneously. We conduct experiments on two representatives of both worlds, DQN (Mnih et al., 2015) for Q-learning methods, and SAC (Haarnoja et al., 2018a) for actor-critic, to test the generality of our method.

### 6.1 ACTOR CRITIC METHODS

Soft Actor-Critic (SAC) is the state-of-the-art algorithm in off-policy actor-critic domain. It focuses on stability and data-efficiency, and even can be applied to challenging real-world robot control (Haarnoja et al., 2018b).

We compare adaptive n-step with fixed n-step and Retrace on the SAC algorithm. For each update target, we use it in place of single-step TD target in original SAC implementation and evaluate on Gym MuJoCo (Brockman et al., 2016) benchmark.

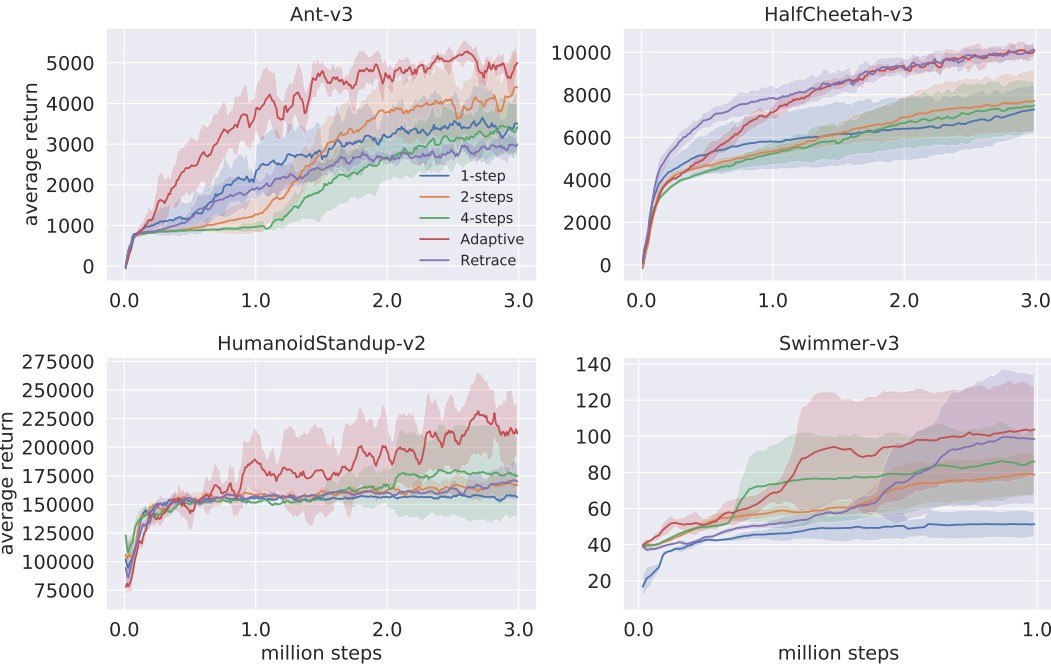

Figure 4: Training curve on continuous control benchmark. We report top 4 in a total 8 runs.

The result shows that adaptive n-step returns outperforms all fixed n-step returns consistently across all tasks. Adaptive n-step also has the lowest variance across different runs, being more stable. It is worth to note that different environments have different best performing fixed $n$, while adaptive n-step perform well with only one set of $n_{max}$ and $d$ across all environments. This suggests that optimal $n$ varies with different data points, while a fixed $n$ is only a coarse approximation.

Adaptive n-step also outperforms the Retrace method, both in terms of average return and computational cost. The calculation of Retrace target requires $2n$ $Q_\pi$ and $3n$ $\pi$ evaluations per step ($n$ is trace length, $n = 32$ for our MuJoCo experiment), and each evaluation is a network forward pass. In contrast, adaptive n-step target inherits $O(1)$ complexity of the vanilla n-step method, only evaluates $1$ $Q_\pi$ and $1$ $\pi$ for arbitrary $n$, thus makes it much faster than Retrace.

## 6.2 Q-LEARNING

DQN (Hessel et al., 2018) sets the foundation of combining Q-Learning with deep neural networks, we pick it as the representation of Q-learning methods. We conduct our experiment on a subset of Atari 2600 games. We compare the adaptive n-step with fixed n-step with $n = 3$, which is the best $n$ on Atari games recommended by Hessel et al. (2018) and Fedus et al. (2020).

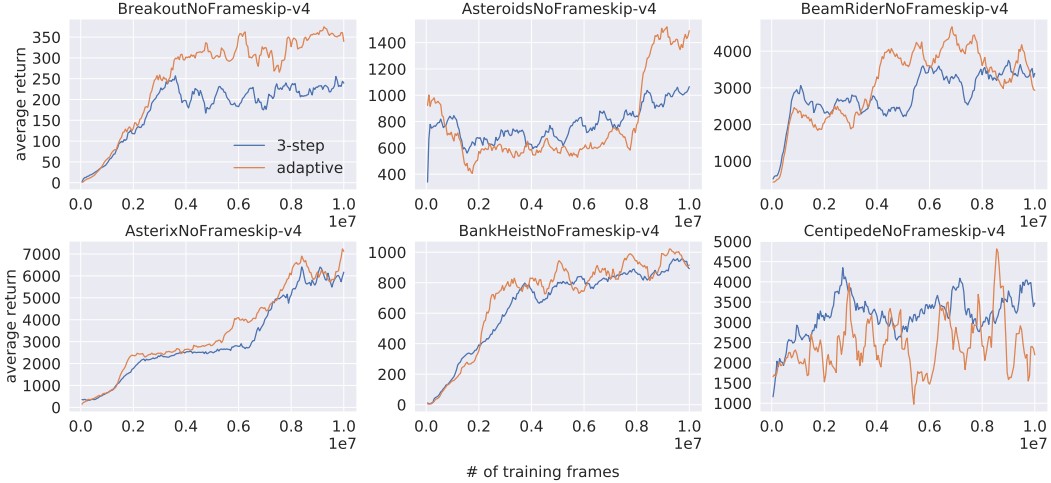

Figure 5: Comparison on Atari 2600 games. For both fixed 3-step and adaptive n-step, we report the agent that obtained highest reward during training. The figure is smoothed by moving average with length 10 to improve readability.

As shown in Figure 5, adaptive n-step outperforms the fixed value n-step returns in all games. And in most of them, the performance benefit brings by the adaptive n-step method exceeds 20%.

## 7 CONCLUSION

Generally, n-step bootstrapping is simply viewed as a unification between Monte-Carlo returns and one-step temporal difference. However, with the introduction of replay buffer to apply reinforcement learning on large-scale problems, we figure out that n-step bootstrapping actually serves as a control factor to reduce the estimation error. Thus, the selection of $n$ should differ on each data point instead of a fixed value.

Based on this observation, we propose the adaptive n-step bootstrapping algorithm to select the value of $n$ for each data point individually. Experimental results show that adaptive n-step outperforms all fixed value $n$ settings with a large margin. Comparing with other update target definitions, e.g. Retrace, adaptive n-step bootstrapping only introduces negligible computation cost and is easy to implement. Those characters make it be easily embedded into other algorithms.

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
