# OpenReview forum: "Adaptive N-step Bootstrapping with Off-policy Data"
_ICLR.cc/2021/Conference — Reject_

### Official Review · AnonReviewer4 · 2020-10-16
**Interesting observations with promising experimental result**

**Rating:** 5
**Confidence:** 4

**Review:**

### Evaluation
Although the paper presents interesting observations and some promising experimental results, I recommend rejection. The proposed method is not general, because it is purely based on empirical observations obtained with only SAC and Q-learning. As a result, it is unclear if the proposed method can be combined with other algorithms. The proposed method is also a kind of hack that makes use of the overestimation problem, and thus, it won't work if we solve the overestimation problem. Besides, the paper is not well-written, and experimental methods are a bit inappropriate.

### Paper Summary
Uncorrected N-step return is very effective and frequently used in off-policy RL algorithms, such as R2D2 and Ape-X. Its drawback is that "N" is determined heuristically or by a hyperparameter search. The paper presents interesting observations that

1. an error between expected uncorrected N-step return and the true Q-value mainly consists of underestimation due to uncorrected off-policy bias and Q-value overestimation due to function approximation error,
2. balance between the underestimation and overestimation can be controlled by "N", and that
3. an optimal $N$, which minimizes the error, obeys $A e^{-Bp}$ with appropriate $A, B \in (0, \infty)$, where $p$ is _policy age_.

Based on these observations, the paper proposes to change and determine $N$ for each $G_t$ by $round(N_{max}^{1 - \min(1, p/d)})$, where $N_{max}$ and $d$ are hyperparameters. Some experiments show performance gain compared to algorithms with fixed N.

### Strong Points
1. Interesting observations on uncorrected N-step return (described in the summary above)
2. Some performance gain in MuJoCo.

### Weak Points
1. The paper is not well-written. It contains typos, unclear sentences, and ambiguous notations.
2. The proposed method for choosing N does not seem to be general. (I feel it is specialized in SAC and Q-learning. Besides, it is a kind of hack that does not work if we solve Q-value overestimation.) I wonder if this method works for other algorithms, especially algorithms using trust region methods.
3. The abstract says a critical issue of uncorrected N-step return is that one needs to choose $N$. The proposed method allows us to avoid choosing $N$, but now we need to choose $N_{max}$ and $d$, which also seems to be difficult.
4. SAC's performance in the paper is substantially lower than that of a publicly available implementation (e.g., [SpinningUp](https://spinningup.openai.com/en/latest/spinningup/bench.html)). This made me a bit skeptical of the SAC implementation of the paper.
5. Figure 4 shows average learning curves of top 4 agents. Figure 5 shows learning curves of best agents. I feel mean, median, and std across all agents should be provided.

### Comments to the Authors

1. The paper is not well-written. Please revise it again. Besides, please revise the references section. It refers to arxiv versions of papers that were accepted to conferences.
2. Is it possible to show that the proposed method can be combined with other algorithms? In particular, I am wondering if the proposed method can be combined with algorithms using trust region methods. (For example, PPO?)
3. SAC's performance in Figure 4 seems to be substantially lower than that of a publicly available implementation (e.g., [SpinningUp](https://spinningup.openai.com/en/latest/spinningup/bench.html)). Would you share your code so that I can check the implementation?
4. Or would you run SAC experiments with SpinningUp using N-step return?
5. Why do you show learning curves of best agents (or top 4 agents) rather than mean or median?

---

### Official Review · AnonReviewer3 · 2020-10-22
**Interesting idea, needs further development**

**Rating:** 4
**Confidence:** 5

**Review:**

This paper proposes an approach to adapting the n parameter in n-step returns according to the off-policyness of the sampled transition. The key novelty to approximate this off-policyness using the age of the policy that generated the data, and demonstrating that there is a fairly reliable relationship between the two in simple experiments.

Overall, I enjoy the approach taken here, and I think it deserves a place in the landscape of deep RL research. However, I'm not sure that the paper is of sufficiently quality to be accepted as-is. The paper is quite roughly presented, with a number of imprecisions and technical inaccuracies. The empirical results are interesting, but would benefit from being substantiated.

Section 4.1. There are a number of issues here which I'm curious to hear your comments on.
1) Equation (2): What is the expectation on mu over? Just a?
2) In this case, where are the r tilde coming from? There seems to be some kind of other sampling
procedure generating this.
3) E_approx(.) is defined in terms of an expectation over mu (c.f. Eqn 2), but I think you really mean pi.
4) As a consequence, the derivation of E_residual is not quite right. I think you end up with a q_pi - q_mu term. Please confirm, and explain what that means for your argument.
5) In this context, what does that mean for Eqn 3? Clearly if gamma is large, the first two terms are small. But what about the q_pi - q_mu term?
6) Actually, the derivation of E_residual seems to imply that the expectation in Eqn 2 is over a full trajectory, otherwise you cannot swap out the sample partial returns with Q-values. But in this case there may be other terms, e.g. the distributions of s_n and s tilde n come from the visitation distribution induced by mu, which is different from that induced by pi.

Section 4.2. "The approximation error is an overestimation ..." I don't understand this paragraph, or why it is necessary from the previous section's derivation. Can you give a formal argument for this point? Clearly Fig. 1 shows that the effect is happening, but I don't think your math supports it.

Your definition of n* (Equation 5) only seems to make sense in deterministic environments. Please comment.

At a higher level, there are a number of presentation gaps between stated results and the theory given. It would be useful to emphasize which of the conclusions are given by theory, versus experiments, versus conjecture.

Fig. 2 suggests an almost linear relationship between p and the log-differences. Is this because you are applying SGD with a step size of 0.001, and hence finding that a first-order Taylor expansion to mu, centered at pi, fits well here?

Fig 3. Why are all the x axes different? Can you explain these graphs -- would they look the same if the problems were stochastic?

"In summary, as the policy age grows, the off-policy bias increases linearly ..." where do you show this? Is this an anecdotal observation from your experiments?

You optimize hyperparameters on one environment... this matters a lot here. Can you give some details?

Fig 4. What hyperparameters are you using in the end?

Fig 5. The figure titles are quite rough. Better would be to use the actual game name, not the Gym environment string identifier. Also, I believe v4 is without sticky actions, which woudl make the domain deterministic. Can you confirm?

In general, the results of fig. 5 don't support the claim that the method is having a significant impact. The claim that the adaptive method outperforms on all games also isn't true -- it's clearly worse on centipede.

Where you cite Thrun and Schwartz, 1999 -- consider citing Gordon, 1995 instead.

---

### Official Review · AnonReviewer2 · 2020-10-26
**Intriguing topic that can benefit from solid theory, which unfortunately does not exist here**

**Rating:** 4
**Confidence:** 4

**Review:**

The paper deals with an intriguing point in RL -- how to correctly choose an adaptive $n$ for $n$-step bootstrap. From such a paper one might expect theoretical results on the tradeoff between bias and variance in the presence of off-policyness. However, as opposed to the picture portrayed in the first section, no such analysis follows.

I believe the research question at hand can highly benefit from rigorous analysis, but other than the rudimentary Eq. (3) and the few equations that precede it there are no established principles according to which one should derive a formula for $n$. Instead, the argument that is given for choosing it regards the 'age' of the policy. But, this argument is hand-wavy and might as well be less important than other factors such as the level of accuracy of the value function per given state at a given time. Also, such a formula for updating $n$ is highly dependent on parameter choice which would naturally change for different environments.

I'm also particularly disturbed by the fact that it seems, by the age heuristic used in the paper, that there's a hidden assumption that DQN is a monotone algorithm. This is even explicitly mentioned in the first sentence of page 2. But it is, of course, incorrect. Based on stochastic approximation analysis, only asymptotic results on convergence exist or finite-time analysis that doesn't assume monotonicity.

There is also a comparison to Retrace [Munos et al., 2016] and claims on the method here being more efficient. I find that selling point acceptable but relatively weak; the Retrace paper is backed by detailed analysis and thorough comparison to other alternatives. A similar distinction can also be made, by the way, to the Vtrace paper [Espeholt et al., 2018].

In terms of experiments, it seems that the proposed algorithm works well. But, as mentioned, I tend to believe it needs tweaking per environment, which is equivalent to tweaking to a 'good' $n$ value.

To summarize, as an empirical paper that focuses on a problem that is very natural to be theoretically analyzed, I consider the contribution here to be not significant enough for publication.

---

### Official Review · AnonReviewer1 · 2020-10-27
**Official Blind Review #1**

**Rating:** 3
**Confidence:** 4

**Review:**

## Summary
This paper provides a novel algorithm to estimate the optimal value of $n$ for $n$-step temporal difference methods. The paper derives an optimal value of $n$ based on minimizing a bias term, then utilizes intuition to derive an online approximation algorithm. The paper compares its adaptive $n$-step algorithm against fixed $n$-step values with both DQN and SAC empirically on several domains.

## Review

### Summary
I am recommending a reject for this paper. The paper is built from the assumption that "the underlying mechanism of why $n=3$ performs better than one-step temporal difference is still unclear." However, the paper misses a discussion on the well-understood bias-variance tradeoff present with $n$-step methods. I do not find that the paper provides any clarity or novelty in the _understanding_ of $n$-step methods. Although the proposed algorithm is novel, I remain unconvinced by the empirical demonstration and the lack of theoretical support for the algorithm.

To increase my rating of the paper, I would have liked to have seen at least some of:
 * A discussion and analysis of the bias/variance properties of the proposed algorithm.
 * A proof that the proposed algorithm has better bias/variance properties than a naive fixed $n$ approach.
 * A statistically sound empirical investigation of the proposed algorithm (even if on smaller domains, like tabular gridworlds).
 * Fewer meta-parameters for the proposed approach.
 * A sensitivity analysis of the meta-parameters for the proposed approach.

### Theory
The theoretical components of the paper were largely poorly written, involving typos, inconsistencies, poorly motivated definitions, several forward references, etc. which made reading and reviewing quite challenging.

* The terms "off-policyness" and "policy age" are used throughout the paper; however, they are only defined intuitively (i.e. not precisely) until section 5 where they are given an unsatisfactory definition. The paper defines "off-policyness" as the number of steps between when the behavior policy was recorded and the current timestep. This, however, does not take into account any aspect of the dynamical system defining the change in policies. For instance, imagine Q-learning with a stepsize of 0. The policy would never change, yet the "off-policyness" by definition would grow linearly according to this definition. The lack of solid definition here requires an unfortunate environment/algorithm specific meta-parameter later in the analysis to account for this fact (more discussion on this later in the review).

* Throughout the paper, the term "error" is used but the formula given is $\mathbb{E}\left[ \hat{G} - v_\pi \right]$ which is the definition of bias. While a minor nitpick in terminology, this leads towards a lot of confusion throughout where some quantities are defined as "errors" while other as "biases", but what is ultimately meant is "bias". The term "approximation error" means something specific generally, and is not used correctly in the paper (for instance). Because "bias" is the only portion of the error that is investigated throughout the paper, then the most important portion of the discussion around $n$-step methods is ignored (namely, the variance). Even simplifying the setting investigated by this paper to being fixed-policy (e.g. policy evaluation) with target and behavior policies the same (i.e. on-policy), then still there is a trade-off between $n=1$ and $n=\infty$. This trade-off results from balancing between bias and variance of the estimated returns. While adding shifting policies and considering the off-policy setting does yield an interesting conversation on bias, the variance simply cannot be ignored.

* The error decomposition is incorrect. It does not account for all of the error in the estimate, resulting in a "residual" unaccounted for error. We do not know what causes this error, we do not know if it implies that the defined accounted for errors are incorrect due to the residual error, or if there is simply some additional term not being accounted for, but we do know that the magnitude of the residual error is non-negligible due to the empirical investigation in Figure 1. Further, because we are actually looking at biases not errors, we cannot make use of the non-negativeness of these quantities to assert their independence. The residual error appears positive. What if the residual error results from not accounting for all of the error due to "off-policyness" (which is always negative) and thus cancels with the off-policy error?

* Where did Equation 7 come from? This appears to be entirely intuited from a couple of empirical plots? Does the equation even make sense? I see $n^* = \text{round}\left( \frac{n_\text{max}}{n} e^{\min(1, \frac{p}{d})} \right)$ which results from an $\text{argmin}_n$. How do we solve this equation with an $n$ on both sides wrapped in a non-linear function? Is this based on Bhandari et al. (2018) where this paper incorrectly reads that gradient descent has an **exponential** convergence rate? Where does gradient descent come into play (TD is not the gradient of any function)? Why limit the exponent to be at least 1? Doesn't this lead to an algorithm where $n=2$ is the smallest n-step method we can consider?

### Empirical Study
I highly appreciated section 4.2. I liked the investigation of the quantities defined in the theoretical section and asserting that assumptions hold. I however do not agree that the empirical results here imply the conclusion drawn. In what way do we see that "N-step returns works because a suitable selection of $n$ makes the overestimation and underestimation cancel each other"? This conclusion is not clear from the results. It also lacks the appropriate nuance given the highly limited study and lack of statistical evidence to support the claim.

How did you choose the meta-parameter value of $d = 122952$? This seems....arbitrary at best. What does the sensitivity of $d$ look like? $d$ appears to result from the fact that you did not account for the rate that the policy is changing, only the number of timesteps that have passed. This makes the fraction $\frac{p}{d}$ become an approximation for policy change rate, where the onus is entirely on the algorithm user to estimate and tune this quantity for their given algorithm, other meta-parameters, and environment.

The rest of the plots in section 6 onwards have a fundamentally broken empirical methodology that renders the results largely uninterpretable. Comparing the mean and standard error (I believe this is what the shaded region shows?) of the top 4 out of 8 runs will not yield statistically valid results. The differences between the algorithms will be lost to variance and noise, making it impossible to know which algorithm is actually best. More runs would be needed and likely a different validation metric as well. Note that using the 50th percentile disproportionately favors high-variance methods as it cuts off their failing runs. This implies to me that the proposed algorithm has much higher variance (as I would guess from the algorithm itself) and only works a small percentage of the time. Perhaps this is still a desirable trait, but this should be stated and investigated explicitly and not hidden behind strange empirical decisions.

I will refer to Henderson et al. 2015 for a discussion on why showing the mean and standard error of the top 4 out of 8 runs is insufficient for a statistically valid methodology. I will also attach a short python script which demonstrates a simplified version of Figure 5 from Henderson et al. 2015, demonstrating very clearly that no conclusions can be drawn from this methodology. Even ignoring my complaint at the unusability of the empirical results in section 6, still the difference in performance of the proposed algorithm and a 2-step baseline appear negligible; especially once one considers the introduction of two new meta-parameters to replace the original one meta-parameter, and that these two meta-parameters appear difficult to tune.

#### A demonstration of methodology
```python
import numpy as np
np.random.seed(0)

N = 8
TOP_N = 4

# hidden underlying process
# let's see which of these two "algorithms" is better
# Note, however, that they are in fact the same
alg1_performance = np.random.normal(0, 0.1, size=N)
alg2_performance = np.random.normal(0, 0.1, size=N)

# only look at top n points
alg1_performance.sort()
alg2_performance.sort()

alg1_top = alg1_performance[-TOP_N:]
alg2_top = alg2_performance[-TOP_N:]

# which is better? (note both should be approximately 0)
print("mean 1:", alg1_top.mean()) # => 0.17
print("mean 2:", alg2_top.mean()) # => 0.08

# can we estimate variance? (Nope)
print("std 1:", alg1_top.std()) # => 0.046
print("std 2:", alg2_top.std()) # => 0.042

# what about confidence intervals? (Nope)
alg1_stderr = alg1_top.std() / np.sqrt(TOP_N)
alg2_stderr = alg2_top.std() / np.sqrt(TOP_N)

print("CI 1:", (alg1_top.mean() - alg1_stderr, alg1_top.mean() + alg1_stderr))
print("CI 2:", (alg2_top.mean() - alg2_stderr, alg2_top.mean() + alg2_stderr))
# yields CI 1: (0.15, 0.19)
# yields CI 2: (0.06, 0.10)
# these confidence intervals don't overlap, therefore
# conclusion: they must be different algorithms
# and alg1 has better average performance, so alg1 must be better
```

---

### Decision · Program_Chairs · 2021-01-07
**Final Decision**

**Decision:**

Reject

**Comment:**

This paper studies n-step returns in off-policy RL and introduces a novel algorithm which adapts the return’s horizon n in function of a notion of policy’s age.
Overall, the reviewers found that the paper presents interesting observations and promising experimental results. However, they also raised concerns in their initial reviews, regarding the clarity of the paper, its theoretical foundations and its positioning (notably regarding the bias/variance tradeoff of uncorrected n-step returns) and parts of the experimental results.
In the absence of rebuttal or revised manuscript from the authors, not much discussion was triggered. Based on the initial reviews, the AC cannot recommend accepting this paper, but the authors are encouraged to pursue this interesting research direction.